# Modular, focused data science education improves biomedical learners' abilities: A study of the Data and Analytics for Research Training (DART) program

Rose Hartman[1]☉*, Karen Joy Payton[1]☉, Rose Franzen[1], Meredith Lee[1], Elizabeth Drellich[1], Ali Shokoufandeh[2], Jeffrey Pennington[1]

1 Department of Biomedical and Health Informatics, Children's Hospital of Philadelphia, Philadelphia, Pennsylvania, United States of America, 2 College of Computing & Informatics, Drexel University, Philadelphia, Pennsylvania, United States of America

☉ These authors contributed equally to this work.
* hartmanr1@chop.edu

## Abstract

The increasing availability of big data and adoption of sophisticated computational techniques in biomedical research has exciting implications for our scientific understanding of human health. However, researchers report struggling to find data science education that meets their needs, despite the fact that many training programs and online resources exist. There is a lack of evidence on the strengths and weaknesses of various training options, making selecting an educational path daunting. We created a new data science training program focused on rigorous, reproducible methods for biomedical research, making use of tightly scoped modular content that can be flexibly arranged to provide a curriculum tailored to a researcher's specific needs and skill level. Moreover, we ran a study testing the program's effectiveness, providing not only another option for data science training but also a model for collecting and sharing relevant data on data science education programs. We ran two waves of research participants, adjusting our materials in between to improve both the training program and our research design. For both waves, we pre-registered hypotheses that learners' self-reported data science ability and level of agreement with important tenets of open science would increase over the course of the program. Indeed, learners showed significant improvement in data science ability (Wave 1: $t(47) = 10.18$, $p < .001$, Wave 2: $t(238) = 17.12$, $p < .001$) and greater agreement with open science values (Wave 1: $t(47) = 3.56$, $p < .001$, Wave 2: $t(238) = 7.95$, $p < .001$). Follow up analyses underscore the robustness of improvement in data science ability. The improvement in open science values was more moderate and was significant only in some of our pre-registered hypothesis tests, likely due to a ceiling effect as most learners reported high agreement with open science values at pretest.

**Data availability statement:** All data and code for this study are available in a GitHub repository at https://github.com/arcus/DART_analysis. We have also used Zenodo to assign a DOI to the repository: https://doi.org/10.5281/zenodo.13323465. Full text .pdf documents of the instruments given to participants are included in our preregistration: https://doi.org/10.17605/OSF.IO/ZMNR6.

**Funding:** JP received funding for this work from the National Institutes of Health (https://www.nih.gov/), grant NIH 5R25GM141501. The funder had no role in study design, data collection and analysis, decision to publish, or preparation of the manuscript.

**Competing interests:** The authors have declared that no competing interests exist.

## Author summary

Data science skills are crucial to the advancement of science, but many working biomedical researchers lack adequate technical training. We built the Data and Analytics for Research Training (DART) program of training modules, and we surveyed participants on their data science skills and attitudes towards open science principles both before and after their participation. We found that the DART program significantly improved learner's ratings of their own technical abilities. We found a less robust change in participants' attitudes towards open science, which was unsurprising as most participants came into the study with existing positive attitudes towards open science.

In addition to demonstrating the effectiveness of the DART program itself, we also present a useful framework for measuring the self-reported impact of educational resources. Our hope is that both this framework and the educational materials we created will be of use to learners, educators, and education researchers.

## Introduction

Researchers seeking to learn data science face a rich selection of options, but simple availability of material is insufficient to make meaningful change in researchers' ability or behavior; funders and researchers alike continue to call for better education in data science. The high demand for data science education has led to a proliferation of learning programs, but has not, to date, been accompanied by robust studies of their effectiveness. Here we introduce a new data science training program alongside a rigorous study of its effectiveness.

### Why data science education?

Deficient statistical rigor and poor research reproducibility are a challenge to the conduct of science and can be exacerbated by inadequate training in data science. In a survey of life science researchers, Attwood et al. [1] found that many respondents reported never receiving any formal training in bioinformatics and data science, and many also reported a lack of confidence in those abilities. Over the last decade, calls for better data science education for researchers have formed a steady undercurrent within biomedical research [2–7].

### Data and Analytics for Research Training Program (DART)

There are many examples of high-quality data science educational options. Data science degree programs, while comprehensive, are time-consuming and may be expensive, and are not closely scoped to the specific needs of working biomedical researchers. Moreover, life sciences researchers generally rate formal degree programs as their least preferred way to acquire new skills [1]. Other options, such as certificate programs and free or inexpensive online course platforms (DataCamp,

Coursera, EdX), provide a rich ecosystem for learners beyond formal degrees. However, scientists still express frustration and discouragement with the current availability of education and a desire for a better way to learn data science [8,9].

Several recent projects have begun to address this gap. One of the best-established projects is the Carpentries organization, which supports workshops in software and data skills for researchers across a range of disciplines around the world [10]. Auer et al. [11] describe a large-scale education program for researchers (R4E), focused primarily on reproducibility. Their audience was life science researchers specifically, and content was designed to be delivered primarily via workshops. Similarly, Chen [12] designed and delivered workshops to build data science skills in a population of health science researchers, but added an additional component of customization, using self-report surveys to construct empirical personae to tailor instruction and contribute to the body of knowledge about the profiles of learners served in programs like this. There is also a growing movement in open data science education, beyond just open science, including publishing educational content freely and licensing it liberally for reuse [10–14].

We likewise saw a need for an effective method of providing research-focused data science skill acquisition, but unlike the excellent programs discussed above, we opted not to rely on synchronous training options, allowing our program to complement rather than duplicate those efforts. Scalability is a perennial concern for workshop-based training programs [15], as is the challenge of how to support long-term growth and learning with short-format synchronous instruction [15,16]. Moreover, scheduling constraints can put synchronous training out of reach for even the most motivated researchers. As data science educators in our institution, our experience suggested that an asynchronous approach, rather than synchronous workshops, might scale more effectively and provide greater reach. We saw an opportunity to take some of the most effective elements of existing trainings – the hands-on focus of Carpentries [10], the learner customization Chen [12] piloted, the inclusion of practical approaches to improve reproducibility championed by R4E [11] – and adapt them for a fully asynchronous format.

Opting for a fully asynchronous learning environment was a tradeoff we weighed carefully. Although asynchronous learning environments offer the potential for increased flexibility, accessibility, and scalability, they also introduce challenges to learner engagement. Research suggests that the lack of real-time interaction in asynchronous learning often leads to feelings of isolation, reducing motivation and diminishing students' connection to the learning process [17]. Moreover, the absence of immediate feedback from instructors or peers may make self-regulation more difficult, increasing cognitive load and leading to frustration [18]. On the other hand, a primary goal for us was combining scalability with the ability to tailor instruction closely to individual learners' needs, precluding a traditional synchronous approach to instruction. We focused our efforts on creating a highly scalable program that would respect the constraints of the working investigator and be directly applicable to the practical needs of conducting research. This led us to create the Data and Analytics for Research Training (DART) program.

The foundation of the DART program is a collection of tightly scoped, just-in-time educational modules with a duration of one hour or less, which are written with biomedical researchers as the intended audience. The modules are designed to be standalone and asynchronous, meaning that learners can complete them independently, focusing only on the skills they need. In this way, DART circumvents the constraints that reduce the effectiveness of current data science training options for researchers. While modules can be completed as solo, standalone units, in the DART program we also include pathways that provide a coherent subset of modules in a given sequence. We begin by analyzing participants' needs assessment responses in R, using a statistical method called hierarchical clustering. This approach groups participants with similar responses into clusters. Once the clusters are formed, we design specific learning pathways that directly align with the interests and aspirations of each cluster. For instance, if a particular cluster demonstrates a strong desire to learn Python, we create a pathway that focuses on acquiring skills in Python. If another cluster indicates interest in a wide range of topics, with relatively little current expertise, we develop a pathway that highlights "demystifying" modules offering a broad orientation to a variety of data science topics. The rationale behind employing these tools is to ensure that our learning pathways are meticulously crafted in a data-driven and personalized manner, catering to the unique requirements

of each group. This targeted approach empowers us to respect the time constraints of busy researchers by only delivering content that is highly relevant to their present needs.

Given the proliferation of training programs without investigation of their efficacy, we were unwilling to introduce a new training program without also rigorously studying its impact. In this paper we introduce the DART program along with a carefully designed, pre-registered study measuring its effectiveness. We used a pre-post design to measure change in data science ability, borrowing Chen's four-point Likert scale of ability [12], as well as measuring agreement with important tenets of open science to estimate the effect of participation in our program on learners. We find strong evidence of improvement in data science ability, indicating that the DART program was a useful educational tool. Less of an effect is seen on open science attitudes, likely due to a ceiling effect. This represents two important contributions to data science education: a proven training program (DART), and a potential model for future studies assessing other programs.

## Results

### Hypotheses

To test the approach of applying tightly scoped, research-focused data science training, we created a set of educational modules and pre-registered our hypotheses [19]. Original pre-registered hypotheses for pre- to post-intervention change included: an increase in self-rated data science ability (Hypothesis 1, Wave 1 and 2), increase in positive attitudes regarding open science (Hypothesis 2, Wave 1 and 2), and greater change in both data science ability and attitudes toward open science for more engaged learners (Hypothesis 3, Wave 1 and 2). Additional hypotheses pre-registered prior to the second wave of participants included greater change for learners who like asynchronous education (Hypothesis 4, Wave 2 only) and greater change for learners who express satisfaction with their customized curriculum (Hypothesis 5, Wave 2 only). Hypothesis testing was conducted independently for each of two waves of participants, due to instrumentation and analysis changes between the waves.

### Findings: Wave 1

We collected participants' (n = 97) ratings of their own current level of ability in a variety of data science skills (e.g., "Build a data processing pipeline that can be used in multiple programs"), as well as their level of agreement with important tenets of open science and reproducibility (e.g., "Open and efficient data sharing is vital to the advancement of the field") before and after the program. The program provided each participant with a given pathway designed for that learner's cluster and its particular combination of needs. 100% of consented participants (n = 97) completed the needs assessment and were assigned pathways; of these, 52% (n = 50) completed both the pre- and post-intervention instruments. As per our preregistration, participants with missing data on any of the analysis variables were dropped (listwise deletion), resulting in 48 complete cases for analysis.

We specified in our preregistration that we would use mixed effects models to test our hypotheses whenever possible (whenever the models would converge), using pathway as a random effect. A linear mixed effects model, also called a multilevel model, is an extension of linear regression, but it is more appropriate for grouped data (in this case, learners grouped by pathway) which would violate the general linear model assumption of independent errors if estimated without taking the grouping structure into account [20]. Mixed effects models are more complex than ordinary least squares regression models, however, and it is sometimes impossible to estimate a mixed effects model on sparse data; when our mixed effects models did not converge, we report linear regression results instead. We also report results from the linear models alongside the mixed effects models to provide a point of comparison for readers more familiar with linear regression. There was strong support for the hypothesis that learners' ability improved over the DART program (Hypothesis 1). A random intercepts model with pathway as random effect showed a significant improvement in participants' self-rated ability on data science tasks from pretest to post, $b = 0.84$, $t(5.22) = 9.69$, $p < .001$ (all coefficient tests for mixed effects models

are reported using Satterthwaite's approximation for degrees of freedom). A paired t-test, ignoring the grouping structure (pathways) altogether, also showed a significant improvement in participants' self-rated ability on data science tasks from pretest to post, $t(47) = 10.18$, $p < .001$ (mean (SD) change is 0.84 (0.57) on a 4-point scale from 1 "I wouldn't know where to start" to 4 "I am confident in my ability to do it"), an effect size of $d = 1.47$ (Fig 1).

The hypothesis that participants' level of agreement with open science values would increase (Hypothesis 2) was not clearly supported by the data, although there was some suggestion of a trend in the predicted direction. The planned random intercepts model was singular and therefore could not be estimated. Our pre-registered contingency plan in this case was to run a linear model with pathway as a fixed effect instead, which showed no significant improvement in agreement with open science values in any of the 8 pathways: $R^2 = .26$, $F(8, 40) = 1.75$, $p = .116$. A paired t-test, ignoring the grouping structure altogether, showed a significant improvement in participants' self-rated level of agreement with open science values from pretest to post, $t(47) = 3.56$, $p < .001$ (mean (SD) change is 0.21 (0.41) on a 7-point scale from 1 "strongly disagree" to 7 "strongly agree"), an effect size of $d = 0.51$. An examination of the raw scores for open science items revealed a probable ceiling effect; the mean open science score at pretest was already 6.25 on a scale from 1 to 7, so there was no room to improve for many participants (Fig 2).

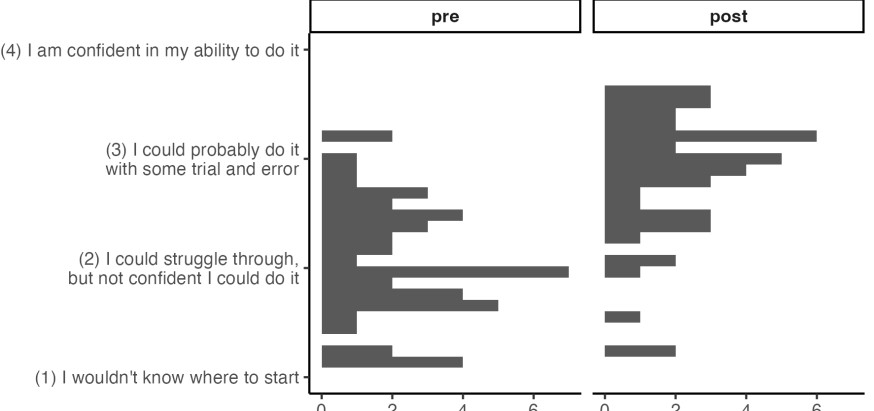

**Fig 1. Data Science Ability Change, Wave 1.** Improved ability (0.84 on 4-point scale) across all data science skills measured.

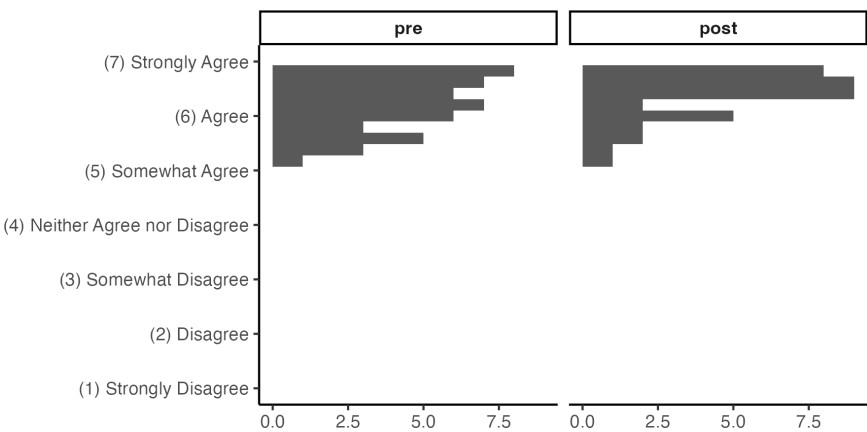

**Fig 2. Open Science Values Change, Wave 1.** A small average increase (0.21 on a 7-point scale), with a ceiling effect noted in pretest scores.

Education was provided via a learning management system that allowed for learner interaction with other learners and the study team. Hypothesis 3, that change in both ability and open science values would be stronger for learners who report higher levels of engagement in the program, was partially supported in the data. Engagement significantly predicted change in self-rated ability ($R^2 = .10$, $F(1, 46) = 4.86$, $p = .033$) but not change in level of agreement with open science values ($R^2 = .00$, $F(1, 46) = 0.18$, $p = .673$), ignoring the grouping by pathways. A linear model with pathway as a fixed effect showed no significant effect of engagement on open science values for any of the pathways ($F(7, 33) = 0.47$, $p = .846$), but there was some evidence of an effect of engagement on change in ability in a model with pathway as a fixed effect ($F(7, 33) = 3.84$, $p = .004$). This result was ambiguous, however, because the significant interaction between engagement and pathway appeared to be driven by respondents in a single pathway ("teal" pathway, $n = 9$), rather than being a more general effect. Mixed effects models including engagement (with random intercepts only, or with random intercepts and random slopes for engagement) did not converge for either the ability or open science outcome.

One possible explanation of the lack of an effect of engagement was the quality of the Wave 1 engagement index itself. It was the average of participants' reported amount of interaction with other participants and amount of completed modules, but overall reported interaction with other participants was very low (mean = 2.08 on a scale of 1 = Never, 2 = Very little, 3 = Sometimes, 4 = Frequently, 5 = Very frequently) and the correlation between completed modules and interaction with other participants was small ($r = .06$). We had expected interaction with other participants and completion of modules to be highly correlated, both indicating an overall level of engagement with the program, but that was not the case. Qualitative feedback from some participants spoke to frustration with a lack of opportunities to connect with other participants, and a lack of response from other participants when interaction was attempted; these barriers to interaction may have undermined the effectiveness of that measure as an indicator of engagement. For Wave 2, we updated our analysis plan to use only the proportion of modules completed as a measure of participants' engagement in the program.

### Findings: Wave 2

For our second wave of participants (n = 422), we broadened the topics covered by our modules, needs assessment, and the pre-post knowledge, skills, and attitudes measure. We added additional pre-registered hypotheses along with analysis plans for the added hypotheses. We updated elements of our exit survey, including altering our measure of engagement to include only module completion level and adding questions about preference for asynchronous education and satisfaction with pathway fit. We changed our learning management system to one that could be administered at no cost, which entailed some changes in the way learners interacted with one another and with study staff. Other study procedures and instruments remained unchanged. 99% of consented participants (n = 419) completed the needs assessment and were assigned pathways; of these, 61% (n = 255) completed both the pre- and post-intervention instruments. As per our preregistration, participants with missing data on any of the analysis variables were dropped (listwise deletion), resulting in 239 complete cases for analysis.

There was strong support for the hypothesis that learners' ability improved over the DART program (Hypothesis 1). A random intercepts model with pathway as random effect showed a significant improvement in participants' self-rated ability on data science tasks from pretest to post, $b = 0.62$, $t(14.87) = 14.38$, $p < .001$. A paired t-test, ignoring the grouping structure altogether, also showed a significant improvement in participants' self-rated ability on data science tasks from pretest to post, $t(238) = 17.12$, $p < .001$, mean (SD) change is 0.62 (0.56) on a 4-point scale from 1 "I wouldn't know where to start" to 4 "I am confident in my ability to do it", an effect size of $d = 1.11$ (Fig 3).

We also saw a significant increase in participants' level of agreement with open science values (Hypothesis 2), $b = 0.26$, $t(15.85) = 7.09$, $p < .001$, but with a more modest effect size than observed with the ability ratings. A paired t-test, ignoring the grouping structure altogether, showed a significant improvement in participants' self-rated level of agreement with open science values from pretest to post, $t(238) = 7.95$, $p < .001$, mean (SD) change is 0.26 (0.5) on a 7-point scale from 1 "strongly disagree" to 7 "strongly agree", an effect size of $d = 0.51$ (Fig 4). An examination of the raw scores for open

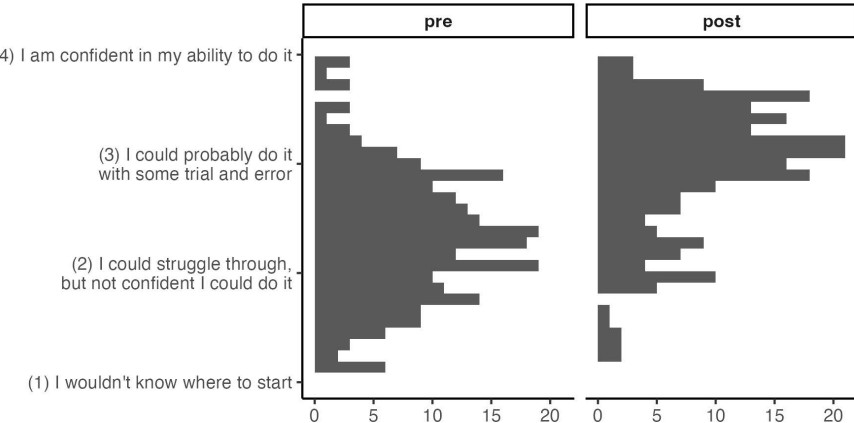

**Fig 3. Data Science Ability Change, Wave 2.** Improved ability (0.62 on 4-point scale) across all data science skills measured.

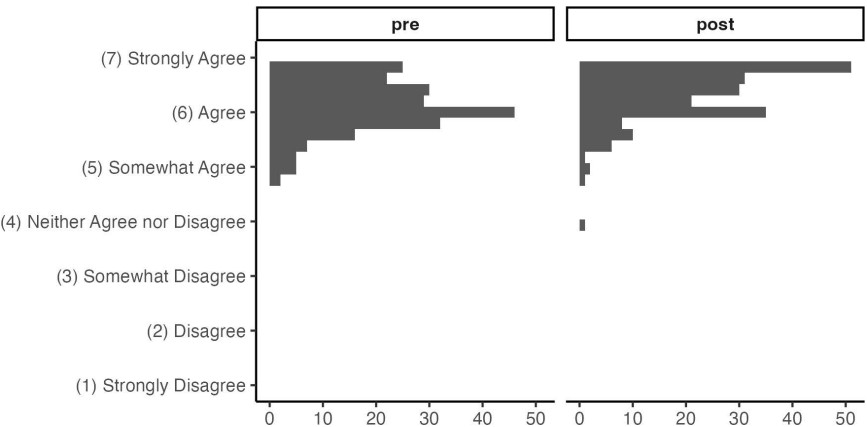

**Fig 4. Open Science Values Change, Wave 2.** Replicated the moderate improvement observed in Wave 1, again with a ceiling effect.

science items revealed a probable ceiling effect; the mean open science score at pretest was already 6.19 on a scale from 1 to 7, so there was no room to improve for many participants, potentially resulting in a muted effect.

We were able to learn more about the nature of the gains observed over the course of participation in DART by controlling for relevant covariates. In particular, if the DART program itself was driving gains, we would expect to see more improvement for participants that engaged more in the program, i.e., completed more of their assigned modules (Hypothesis 3). We tested the predictive power of engagement by re-running the mixed effects models above with engagement added as a predictor. Our preregistration specified that we also allow a random slope for each covariate, with the backup plan that if the random slopes model failed to converge we would revert to a random intercept only; we were able to include a random slope for engagement in the model predicting change in ability, but only a random intercept in the model predicting change in open science values. In the mixed effects models, percent of assigned modules completed ("engagement") did not significantly predict change in ability from pre to post, $b = 0.28$, $t(13.22) = 1.98$, $p = .068$, nor change in agreement with open science values, $b = 0.11$, $t(235.24) = 1.20$, $p = .230$. Ignoring the grouping structure (pathways), there was a significant positive relationship between engagement and change in ability ($b = 0.26$, $t(237) = 2.49$, $p = .014$) but still no change in open science values ($b = 0.11$, $t(237) = 1.20$, $p = .232$), Fig 5.

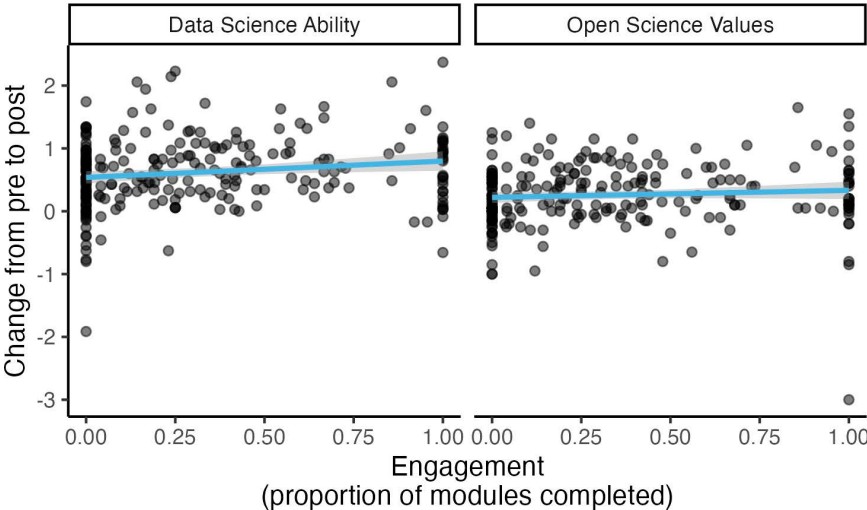

**Fig 5. Engagement, Wave 2.** Overall, the proportion of modules completed is associated with data science skills gains but does not have a significant effect on open science values.

Similarly, we might have expected that a program like DART would work better for learners who reported generally doing well with asynchronous education; their preference might have allowed them to get more from the program, increasing its effect. Indeed, the degree to which participants agreed with the statement "Self-paced asynchronous studying works well for me in general" significantly predicted change in ability, $b = 0.07$, $t(235.18) = 2.10$, $p = .037$ (without accounting for grouping by pathway, $b = 0.07$, $t(237) = 2.13$, $p = .034$). Preference for asynchronous learning did not significantly predict change in open science values, though, $b = 0.03$, $t(235.35) = 1.09$, $p = .277$ (without accounting for grouping by pathway, $b = 0.03$, $t(237) = 1.02$, $p = .309$), Fig 6.

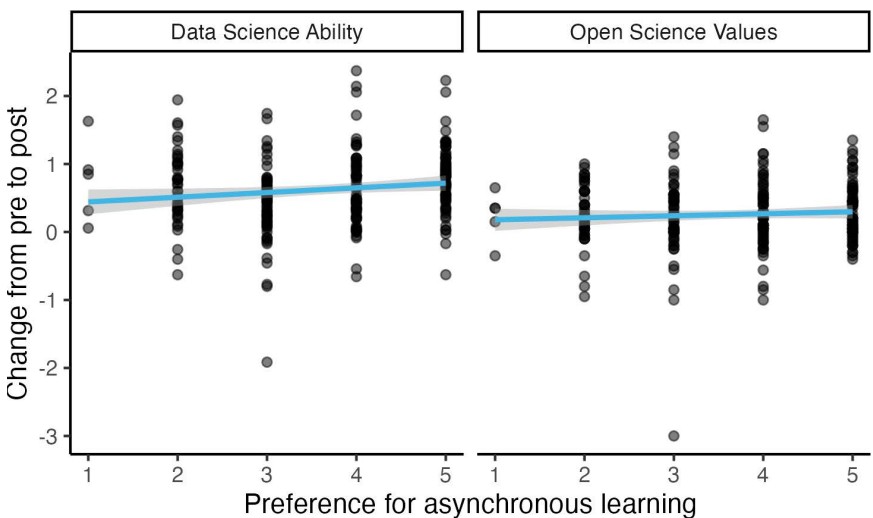

**Fig 6. Preference for Asynchronous Education, Wave 2.** Overall, increased preference for asynchronous training is associated with data science skills gains but does not have a significant effect on open science values.

PLOS Computational Biology

And finally, we predicted that our success in designing an appropriate pathway for each learner would impact how much they gained from DART (Hypothesis 5). We asked a number of questions probing how well their assigned pathway of modules met their needs and expectations and used the mean of their responses on those items to create an index of pathway fit. As predicted, pathway fit significantly predicted change in ability, $b = 0.28$, $t(234.84) = 5.64$, $p < .001$ (without accounting for grouping by pathway, $b = 0.28$, $t(237) = 5.68$, $p < .001$). Pathway fit did not significantly predict change in open science values, though, $b = 0.07$, $t(234.67) = 1.57$, $p = .118$ (without accounting for grouping by pathway, $b = 0.07$, $t(237) = 1.50$, $p = .134$), Fig 7.

Taken together, our results were consistent with the conclusion that DART participants improved in their data science skills because of their participation. Although we did also note an improvement in open science values from pre to post, that change appeared to be independent of participants' experience with DART itself. This may have been the result of an overall increasing commitment to open science in the pool of participants we recruited from in response to changing expectations in their fields, their own growth as researchers, etc.

Of note is the fact that while all participants were assessed on the same set of data science skills and open science values, the actual content of individual learners' pathways differed, and for any given participant many of the measured skills would not have been explicitly covered in their modules. Although more targeted measurement (i.e., assessing just those skills we have explicitly taught) may have resulted in more dramatic gains, an important component of our pedagogical design for DART was building meta-cognitive and psycho-social skills, such as resilience to failure and reduced impostor syndrome. It was our hope that DART participants would come away with an improved ability to learn new data science skills, including on topics we did not cover in their assigned pathway – an ability that should serve them well throughout their career in a rapidly evolving field. Seeing our participants report dramatic gains in self-reported data science skills across such a wide range of topics suggests that DART was effective.

## Discussion

One limitation of the current study was our reliance on exclusively self-report data. There is a large body of literature questioning the validity of self-report assessment; of particular relevance for the current study are the possibility of participants' shifting interpretation of the questions over time [21], and participants' potentially imperfect insight into their own abilities

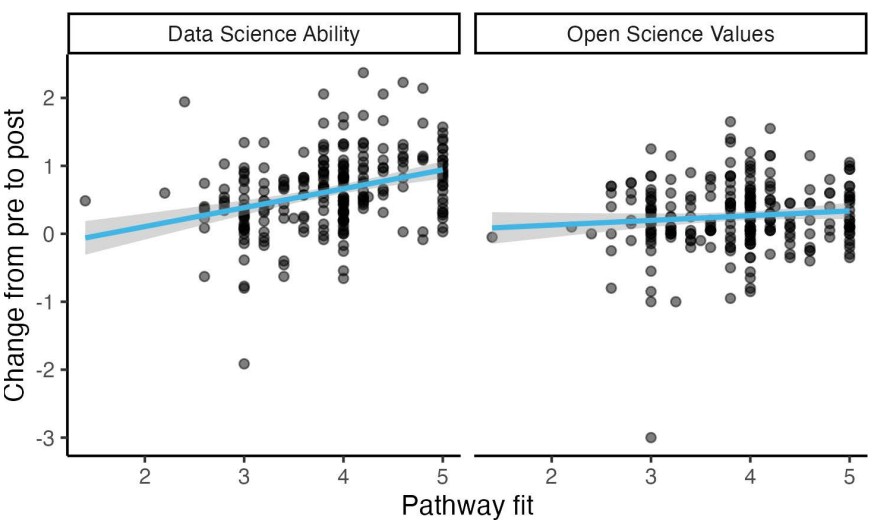

**Fig 7. Pathway Fit, Wave 2.** Overall, increased agreement that the supplied pathway met learner needs is associated with data science skills gains but does not have a significant effect on open science values.

[22]. In other words, the increases we observed in self-reported ability may not actually reflect corresponding increases in participants' objective skill and knowledge. In order to incorporate objective measures of learning in our design, however, we would have needed to sacrifice the flexibility and scalability of our current approach. Self-assessed skill level is also in itself a valuable measurement in education research, as students' evaluations of their own ability may influence important constructs like motivation [23]. The current study shows a clear increase in self-reported ability over the course of participation in the DART program, leaving objective assessment of skill as a promising future direction.

Another important limitation of this study is the self-selecting nature of participants. These results could not be expected to replicate in a population of learners who were assigned DART modules as compulsory education. As educators, we feel there is a clear need for training and support among eager learners; we are content to leave the question of how to motivate researchers who do not already wish to acquire these skills to another study.

Furthermore, we noted substantial attrition in both waves of the DART program, with roughly half of our participants failing to complete the post-test surveys. This level of attrition is typical of online learning programs (e.g., Hadavand et al. [24] report 49% attrition between enrollment and completion in a large sample of online data science learners), but it still raises important questions about the generalizability of the results we observe. Learners who completed the program almost certainly differed from those who didn't in important ways, both measurable and unmeasurable, such as motivation, availability of protected time for study, support from outside of the DART program (e.g., high-quality mentorship), and more. This limits the extent to which we can expect these results to generalize. Our findings are specific to the population who completed the program; we are unable to extrapolate from these data to predict how effective a program like DART might be in a broader population, including the population who enrolled in but did not finish our program.

Problems of attrition and self-selection bias are endemic in online education research, and improving understanding of factors that lead to attrition is an active line of inquiry [25,26]. As with many online learning programs, part of our goal was to lower barriers to enrollment – we made the program free, with no prerequisites, advertised it widely, and created fully asynchronous instruction to allow for maximum flexibility in busy learners' schedules. A likely consequence of this approach is that a high proportion of people signed up without actually having the time or bandwidth to follow through on their learning goals (consider this in contrast to something like a master's program in data science, where requiring a substantial upfront investment from learners results in more selective enrollment, and less attrition). This effect is reflected in the engagement data we were able to capture in Wave 2 after switching platforms; of the 419 enrolled learners, only 243 (58%) logged any activity in their learning pathways after the first day. In other words, fully 42% of our Wave 2 learners never engaged at all with their learning pathways, suggesting that they either changed their minds about participating between enrollment and beginning their learning pathways, or that they maintained an intention to participate but never found the time to do so. We also expect that our choice to rely on asynchronous learning contributed to the attrition we observed; compared to synchronous instruction, asynchronous education has been shown to negatively impact learners' sense of engagement, connection, and depth of learning as well as their motivation [17,18,27–30]. We limited ourselves to fully asynchronous instruction to prioritize scalability, customizability, and the flexibility to accommodate researchers' busy schedules. However, we are excited by the potential for the use of DART materials [31] alongside synchronous instruction; indeed, our team has already successfully incorporated several DART modules into workshops at our institution and are aware of the use of DART modules in synchronous educational efforts in at least one other country as well. In addition to potentially providing content for live instruction, modules offer the opportunity for practice and reinforcement after a workshop, supporting learners with valuable spaced instruction that may otherwise be out of reach in workshop settings [16].

A final important consideration is the question of causality. All participants received access to the training program; we did not hold out a random control group for comparison. This limits the confidence with which we can attribute the gains we measured to DART directly; our participants may have been learning these skills independently, either through another educational platform or through their own growing experience while working on their research. Indeed, we expect that was frequently the case. To provide additional context on the effectiveness of DART *per se*, we can look to the results

that measure the effect of engagement, asynchronous preference, and pathway fit – each of those predicts larger gains in ability. Our exit survey also included questions asking participants directly about how well they thought DART worked. The results underscore the interpretation that learners found DART useful: 87% of participants said they would continue studying with the DART materials even after the end of the program. Future researchers may find other ways to re-use the educational modules we created for the DART program to measure data science training effectiveness using different instructional methods or experimental design, such as a randomized control trial, as these modules are liberally licensed to promote reuse, continue to be updated, and are publicly available [23].

## Materials and methods

### Ethics statement

This research was reviewed by the Children's Hospital of Philadelphia Research Institute Institutional Review Board (IRB # 22–020087) and was determined to meet exemption criteria as educational research per 45 CFR 46.104(d) 2(ii). We obtained written consent from each participant upon enrollment in the program (consent form available on OSF: https://osf.io/j24ku).

### Recruitment

Research participants were recruited in two waves, one for participation beginning in January 2023 and one for participation beginning in August 2023. Investigators sent invitation emails to offices of postdoctoral affairs and faculty development or similar bodies at a number of research institutions in the United States and by using departmental newsletters within Children's Hospital of Philadelphia. Some interested participants independently went on to advertise the study to their colleagues within and beyond their institutions, and the study began with 97 consented participants from 23 institutions in our first wave and 422 participants from 58 institutions in our second wave.

### Materials

Data science educators and subject matter experts created training modules on topics intended to closely mirror the objectives and strategies found in the 2017 NIH Strategic Plan for Data Science [32]. These modules were designed to be asynchronous, brief (consisting of one hour or less of engagement per module), modular (they could be mixed and matched and could stand alone independently without dependencies on other modules), friendly (with encouraging language and metacognitive supports), and accessible (written with inclusiveness standards at the core of the project). We used open source and no-cost methods for developing these modules, intending a low barrier of entry for their production and developing a format that is easy to update. In the interest of maximal sustainability and reproducibility, modules were written with a permissive license for reuse, in a plaintext format known as Markdown, published in a public GitHub repository, and rendered to an attractive, paginated presentation style for learners via a free, open source tool known as LiaScript [33].

### Program

Study data were collected and managed using REDCap electronic data capture tools hosted at Children's Hospital of Philadelphia [34,35]. Before beginning the educational program, participants completed a needs assessment. Similar to the work of Federer et al. [36], this instrument assessed participants' current expertise and professional relevance on a number of data science tasks and techniques, such as analysis of electronic health records, data visualization, and use of programming languages like R and Python. We added an additional measure for each task and technique, asking participants to rate their willingness to learn the topic. These responses were then analyzed using hierarchical clustering – a method where each participant is initially treated as their own group and are gradually merged with others based on

shared characteristics – to create a desired number of roughly homogeneous groups. This process enabled us to identify natural groupings of learners with similar needs. Expert data science educators then manually reviewed these clusters and assigned customized learning pathways to each group. Each pathway consisted of approximately 10 estimated hours of content for Wave 1 or 15 hours for Wave 2. These pathways were designed to address the unique combination of needs and interests discovered within each cluster. After completing Wave 1 and developing additional educational modules, the needs assessment underwent an update by incorporating new questions and modifying or removing others. This ensured that our evaluation tools remained aligned with the most up-to-date and relevant educational materials we provide.

An outcomes survey assessed knowledge, skills and attitudes related to rigorous and reproducible data science and was administered both before and after the program. We pre-registered hypotheses including that participants would rate their skills and attitudes around rigorous and reproducible data science to be higher at the end of the program [19].

In addition to the outcomes survey, participants were asked to complete an exit survey at the end of the program designed to help us identify possible areas for improvement in the materials and administration of the program, especially as relates to issues of accessibility and inclusion. There is also an optional, anonymous feedback form available at the end of each educational module for participants to provide ratings and comments related to that module specifically in order to help us improve the materials. After Wave 1, we added additional items to the exit survey to measure participants' engagement in the program, preference for asynchronous learning, and satisfaction with the fit of their assigned learning pathway.

## Acknowledgments

We are indebted to the Academic Training and Outreach Program at the Children's Hospital of Philadelphia, particularly Wendy Williams and David Taylor, for their support in providing us with pilot (pre Wave 1) postdoc participants. These postdocs helped make this project what it is today, and we are grateful for their generous feedback. The authors would like to thank André Dietrich and Sebastian Zug, the developers of LiaScript. Many of our colleagues on the Arcus team at Children's Hospital of Philadelphia contributed to this project, including Ene Belleh, Peter Camacho, Nicole Feldman, and Colleen Gaynor. Additional thanks go to our interns: Oluwadamilare Agoro, Megan Brown, and Joselinne Piedras-Sarabia.

## Author contributions

**Conceptualization:** Karen Joy Payton, Ali Shokoufandeh, Jeffrey Pennington.

**Data curation:** Rose Hartman, Karen Joy Payton.

**Formal analysis:** Rose Hartman.

**Funding acquisition:** Jeffrey Pennington.

**Investigation:** Rose Hartman, Karen Joy Payton, Rose Franzen, Meredith Lee, Elizabeth Drellich.

**Methodology:** Rose Hartman, Karen Joy Payton, Rose Franzen, Ali Shokoufandeh, Jeffrey Pennington.

**Project administration:** Karen Joy Payton, Rose Franzen, Meredith Lee, Elizabeth Drellich.

**Resources:** Rose Hartman, Karen Joy Payton, Meredith Lee, Elizabeth Drellich.

**Software:** Rose Hartman, Karen Joy Payton, Rose Franzen, Meredith Lee, Elizabeth Drellich.

**Supervision:** Karen Joy Payton, Ali Shokoufandeh, Jeffrey Pennington.

**Validation:** Rose Hartman.

**Visualization:** Rose Hartman.

**Writing – original draft:** Rose Hartman, Karen Joy Payton, Rose Franzen, Meredith Lee, Elizabeth Drellich.

**Writing – review & editing:** Rose Hartman, Karen Joy Payton, Rose Franzen, Meredith Lee, Elizabeth Drellich.

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
