## [Decision Letter · Decision Letter 0]

PCOMPBIOL-D-24-02043

Modular data science education, tightly scoped for biomedicine, improves learners’ technical abilities: A study of the Data and Analytics for Research Training (DART) Program

PLOS Computational Biology

Dear Dr. Hartman,

Thank you for submitting your manuscript to PLOS Computational Biology. After careful consideration, we feel that it has merit but does not fully meet PLOS Computational Biology's publication criteria as it currently stands. Therefore, we invite you to submit a revised version of the manuscript that addresses the points raised during the review process.

Both reviewers agree on the manuscript's methodological soundness and its contribution to data science education for biomedical researchers. The reliance on self-reported outcomes for skill improvement is a limitation that should be better acknowledged. Authors also need to address the high participant attrition (nearly half not completing both pre- and post-tests) as this introduces potential bias and should be discussed more thoroughly.

Please submit your revised manuscript within 30 days Mar 14 2025 11:59PM. If you will need more time than this to complete your revisions, please reply to this message or contact the journal office at ploscompbiol@plos.org. Please include the following items when submitting your revised manuscript:

We look forward to receiving your revised manuscript.

Kind regards,

BF Francis Ouellette

Section Editor

PLOS Computational Biology 

Patricia M Palagi

Section Editor

PLOS Computational Biology 

**Journal Requirements:**

1) Please upload all main figures as separate Figure files in .tif or .eps format. For more information about how to convert and format your figure files please see our guidelines: 

2) Please amend your detailed Financial Disclosure statement. This is published with the article. It must therefore be completed in full sentences and contain the exact wording you wish to be published.

1) Please state: "The funders had no role in study design, data collection and analysis, decision to publish, or preparation of the manuscript."

**Reviewers' comments:**

Reviewer's Responses to Questions

**Comments to the Authors:**

**Please note that one of the reviews is uploaded as an attachment.**

Reviewer #1: the review is uploaded.

Reviewer #2: Thank you for the opportunity to review this paper. It is well-written and methodologically sound and is a useful addition to the literature on data science education for biomedical researchers. My concern about the methodology and discussion of results is that the authors do not discuss the fact that a significant number of participants in both waves, nearly half, didn't complete both the pre and post test. This may introduce bias into the results, in a similar way to when patients are lost to follow up in a clinical trial. We can't know for sure anything about the experience of the people who didn't respond to the post test. Did they find the training useful but just not get around to the post test, or did they stop participating because they didn't see value in the training? It seems reasonable to expect that the participants who did take the time to complete the post test may be different from those who did not in how useful they found the training, and this potential bias should at the very least be addressed in the discussion of limitations.

**Have the authors made all data and (if applicable) computational code underlying the findings in their manuscript fully available?**

Reviewer #1: Yes

Reviewer #2: Yes

PLOS authors have the option to publish the peer review history of their article (what does this mean? ). If published, this will include your full peer review and any attached files.

**Do you want your identity to be public for this peer review?** For information about this choice, including consent withdrawal, please see our Privacy Policy .

Reviewer #1: No

Reviewer #2: **Yes: ** Lisa Federer

**Figure resubmission:**
---

## [Decision Letter · Decision Letter 1]

PCOMPBIOL-D-24-02043R1

Modular data science education, tightly scoped for biomedicine, improves learners’ technical abilities: A study of the Data and Analytics for Research Training (DART) Program

PLOS Computational Biology

Dear Dr. Hartman,

Thank you for submitting your manuscript to PLOS Computational Biology. After careful consideration, we feel that it has merit but does not fully meet PLOS Computational Biology's publication criteria as it currently stands. Therefore, we invite you to submit a revised version of the manuscript that addresses the points raised during the review process.

Please submit your revised manuscript within 30 days Jun 10 2025 11:59PM. If you will need more time than this to complete your revisions, please reply to this message or contact the journal office at ploscompbiol@plos.org. Please include the following items when submitting your revised manuscript:

We look forward to receiving your revised manuscript.

Kind regards,

@bffo

B.F. Francis Ouellette

Section Editor

PLOS Computational Biology

**Additional Editor Comments :**

I fully support the reviewer’s thoughtful suggestions. Implementing these small but important changes — by refining the title, adding examples of high-quality educational options, and rebalancing the discussion to better showcase the study’s strengths — will further enhance the clarity and impact of this already valuable contribution.

**Reviewers' comments:**

Reviewer's Responses to Questions

Reviewer #1: While the study title is proper and descriptive, authors should consider shortening the title.

The authors mentions "There are many examples of high-quality data science educational options." it would be nice if they can provide these examples.

The discussion section needs revision. Currently, it places too much emphasis on limitations, which overshadows the core purpose of this section — that is, to interpret, contextualize, and critically analyze the findings of the study.

Reviewer #2: I appreciate the authors' response to the reviewer suggestions and feel that all of my concerns have been addressed.

**Have the authors made all data and (if applicable) computational code underlying the findings in their manuscript fully available?**

Reviewer #1: Yes

Reviewer #2: Yes

PLOS authors have the option to publish the peer review history of their article (what does this mean? ). If published, this will include your full peer review and any attached files.

**Do you want your identity to be public for this peer review?** For information about this choice, including consent withdrawal, please see our Privacy Policy .

Reviewer #1: No

Reviewer #2: No

**Figure resubmission:**
---

## [Decision Letter · Decision Letter 2]

Dear Dr. Hartman,

We are pleased to inform you that your manuscript 'Modular, focused data science education improves biomedical learners’ abilities: A study of the Data and Analytics for Research Training (DART) Program' has been provisionally accepted for publication in PLOS Computational Biology.

Best regards,

@bffo

--

BF Francis Ouellette & Patricia M Palagi

Section Editors

PLOS Computational Biology

Reviewer's Responses to Questions

**Comments to the Authors:**

Reviewer #1: No further comments. All previous comments and recommendations have been addressed

Reviewer #2: Thank you for the thoughtful revisions.

**Have the authors made all data and (if applicable) computational code underlying the findings in their manuscript fully available?**

Reviewer #1: Yes

Reviewer #2: None

PLOS authors have the option to publish the peer review history of their article (what does this mean? ). If published, this will include your full peer review and any attached files.

**Do you want your identity to be public for this peer review?** For information about this choice, including consent withdrawal, please see our Privacy Policy .

Reviewer #1: No

Reviewer #2: No

---

## [Editor Report · Acceptance letter]

PCOMPBIOL-D-24-02043R2

Modular, focused data science education improves biomedical learners’ abilities: A study of the Data and Analytics for Research Training (DART) Program

Dear Dr Hartman,

I am pleased to inform you that your manuscript has been formally accepted for publication in PLOS Computational Biology. Your manuscript is now with our production department and you will be notified of the publication date in due course.

With kind regards,

Judit Kozma
